# Gut Microbiota in Diagnosis, Therapy and Prognosis of Cholangiocarcinoma and Gallbladder Carcinoma—A Scoping Review

**DOI:** 10.3390/microorganisms11092363

**Published:** 2023-09-21

**Authors:** Ann-Kathrin Lederer, Hannah Rasel, Eva Kohnert, Clemens Kreutz, Roman Huber, Mohamed Tarek Badr, Patricia K. E. Dellweg, Fabian Bartsch, Hauke Lang

**Affiliations:** 1Department of General, Visceral and Transplant Surgery, University Medical Center, Johannes Gutenberg University, 55131 Mainz, Germany; 2Center for Complementary Medicine, Department of Medicine II, Medical Center—University of Freiburg, Faculty of Medicine, University of Freiburg, 79106 Freiburg, Germany; 3Institute of Medical Biometry and Statistics (IMBI), Faculty of Medicine and Medical Center, University of Freiburg, 79104 Freiburg, Germany; 4Institute of Medical Microbiology and Hygiene, Medical Center—University of Freiburg, Faculty of Medicine, University of Freiburg, 79104 Freiburg, Germany; mohamed.tarek.badr@uniklinik-freiburg.de

**Keywords:** cholangiocarcinoma, gastrointestinal microbiome, liver surgery, Klatskin, biliary tract cancer, bile duct, diagnosis, treatment, immune checkpoint inhibitors, mycobiota, fungiom, survival

## Abstract

Cancers of the biliary tract are more common in Asia than in Europe, but are highly lethal due to delayed diagnosis and aggressive tumor biology. Since the biliary tract is in direct contact with the gut via the enterohepatic circulation, this suggests a potential role of gut microbiota, but to date, the role of gut microbiota in biliary tract cancers has not been elucidated. This scoping review compiles recent data on the associations between the gut microbiota and diagnosis, progression and prognosis of biliary tract cancer patients. Systematic review of the literature yielded 154 results, of which 12 studies and one systematic review were eligible for evaluation. The analyses of microbiota diversity indices were inconsistent across the included studies. In-depth analyses revealed differences between gut microbiota of biliary tract cancer patients and healthy controls, but without a clear tendency towards particular species in the studies. Additionally, most of the studies showed methodological flaws, for example non-controlling of factors that affect gut microbiota. At the current stage, there is a lack of evidence to support a general utility of gut microbiota diagnostics in biliary tract cancers. Therefore, no recommendation can be made at this time to include gut microbiota analyses in the management of biliary tract cancer patients.

## 1. Introduction

Biliary tract cancer (BTC) is an overall rare, but highly lethal cancer entity [1,2]. Globally, BTC is much more common in Asia with incidence rates of more than 80 persons per 100,000 per year, e.g., in Thailand compared to an incidence of approximately 1–2 person per 100,000 in Western countries [3]. Recent epidemiological studies have shown a general increase in incidence over the last 20 years [4,5]. The occurrence of BTC is associated with a variety of liver- and biliary tract-damaging diseases such as gall- and bile duct stones and recurrent cholangitis, hepatitis B or C infection, primary sclerosing cholangitis (PSC), parasite infection (especially of *Opisthorchis viverrini*) as well as all kinds of chronic liver diseases such as alcoholic and non-alcoholic liver diseases [2,6,7]. Risk factors are also not evenly distributed worldwide. In Western countries, BTC is more common in the elderly (>65 years of age) and in men [2,8].

### 1.1. Systematic of Biliary Tract Cancers

The biliary tract (also called biliary tree or biliary system) is the generic term for the biliary ducts, the gallbladder and the common hepatic duct. Due to the anatomy of the biliary tract, the term “biliary tract cancers” summarizes a heterogeneous group of tumors. Carcinomas of the biliary ducts and the common hepatic duct are named cholangiocarcinoma (CCA) referring to its origin, the cholangiocytes, specialized epithelial cells lining the biliary tract [9]. More than 90% of CCA are mucin-producing, well-differentiated adenocarcinomas [2]. CCA can occur in all parts of the biliary tract, and a distinction is made between intrahepatic CCA and extrahepatic CCA (perihilar or distal; see Figure 1) [3,9,10]. The different types of CCA share molecular patterns that are relevant to systemic therapy, but the distinction between intrahepatic, perihilar and distal CCA is of high relevance as surgical treatment and prognosis differs significantly [11]. Gallbladder carcinomas are usually named separately from other CCA due to their particularly aggressive tumor biology [12]. Furthermore, unlike CCA, gallbladder carcinoma occurs more frequently in women than in men [8,13].

### 1.2. Diagnosis, Therapy and Prognosis of Biliary Tract Cancers

Early clinical manifestation of all BTCs is unspecific or even asymptomatic [2]. Advanced-stage disease goes along with jaundice, cholestasis and cholangitis depending on the tumor site (Figure 1) [2,14]. Several publications emphasize the necessity to expect BTC in patients with liver and biliary tract diseases, but structured concepts for early detection of CCA and gallbladder carcinoma are widely lacking [9,14,15].

Therapy decision is usually made multidisciplinary and mostly combines hepatobiliary surgery and (adjuvant) systemic chemotherapy in case of resectability [3,16,17,18]. The only potential curative therapeutic approach is the complete surgical tumor resection, but research indicates that curative-intent surgery is only possible in approximately a third of patients [19,20,21]. Because tumor resection requires major hepatobiliary surgery, surgery is associated with a significant risk of complications bearing also the possibility of liver failure or other severe postoperative complications [22,23]. Even though surgery is the only approach to cure CCA and gallbladder carcinoma, long-term survival rates after surgery are still poor, mainly due to rapid disease recurrence [13,17,19]. Recent research indicates that more than one in two CCA patients develop recurrence after curative-intent surgery [24,25,26,27].

### 1.3. The Gut–Liver Axis

Even more than most of the other organs, the liver is in direct contact with the gut microbiota due to the enterohepatic circulation. The portal vein transports gut-derived metabolites to the liver [28]. In turn, the liver is able to “communicate” with the gut microbiota by secretion of bile acids and other molecules which shape the microbial environment (Figure 2).

In 2022, Binda et al. recapitulated three mechanisms of cancer-inducing gut microbiota activity: first, bacterial toxins and metabolites; second, modulation of the host’s local and systemic immune response; third, metabolic changes in the microbiota and the host [29]. The gut–liver axis has a bidirectional relationship that allows communication between liver cells and bile duct cells and the gut microbiota. Therefore, it can be assumed that the mechanisms described by Binda et al. can also play a role in promotion and progression of liver cancer and BTC. Several publications emphasize the role of gut microbiota in in liver diseases and in carcinogenesis. The development of PSC in patients with ulcerative colitis might be provoked by gut microbiota dysbiosis [30,31,32,33]. Furthermore, it has been suggested that gut microbiota dysbiosis might be also able to promote carcinogenesis [34,35,36]. As early as 2012, it was suggested that *Escherichia coli* and its metabolites may contribute to inflammation that promotes the development of colorectal cancer [37]. A potent genotoxin of certain strains of *Escherichia coli*, named colibactin, is able to induce DNA damage in human cells [38]. In 2013, Yoshimoto et al. were able to show that obesity-associated changes in gut microbiota contribute to an increase in intestinal deoxycholic acid, which is known to cause DNA damage. By the enterohepatic circulation, deoxycholic acids provokes a pro-inflammatory type of hepatic stellate cells, which contribute to development of liver cancer in mice [39]. Dapito et al. showed that the toll-like receptor-driven promotion of hepatocellular carcinoma (HCC) in mice is affected by ligands of the intestinal microbiota [40]. Several publications were able to promote data about the role of gut microbiota in HCC genesis, but the evidence for CCA and gallbladder carcinoma is still not clarified. The role of gut microbiota in CCA and gallbladder carcinoma is plausible, but recommendations for considering gut microbiota in BTC patients are still lacking. Therefore, this scoping review aimed to evaluate recent publications regarding the role of the gut microbiota in diagnosis, treatment and prognosis of CCA and gallbladder carcinoma to estimate potential recommendations for gut microbiota handling in BTC patients.

## 2. Methods

We performed a scoping literature review searching databases of Medline, Cochrane Library and Web of Science from database inception to June 2023. The search terms were “‘biliary cancer’ AND gut microbio*”, “‘biliary carcinoma’ AND gut microbio*”, “‘biliary tract cancer’ AND gut microbio*”, “‘bile duct cancer’ AND gut microbio*”, “cholangiocarcinoma AND gut microbio*”, “‘gallbladder cancer’ AND gut microbio*” and “‘gallbladder carcinoma’ AND gut microbio*”. Only English and German abstracts and manuscripts were evaluable. All types of publications were considered, including human and animal trials, with the exception of conference abstracts, editorials, and methodical publications. We checked the reference lists of all relevant manuscripts for other appropriate publications. If the results of reviews were a repetition of original research already included, the reviews were excluded. To ensure quality of this narrative review, the manuscript was prepared according to the PRISMA Extension for Scoping Reviews and the recommendations of von Elm et al. [41,42].

### 2.1. Population

All kinds of animal trials as well as adult patients diagnosed with CCA or gallbladder carcinoma, regardless of gender and curative or palliative intention to treat, were eligible for review inclusion. For review evaluation, it was obligatory that publications provided data about the patient composition of gut microbiota, measured by sequencing or conventional stool cultivation. Since the review focused on gut microbiota, data about tissue or bile microbiota were not considered.

### 2.2. Research Questions

The aim of this scoping review was to evaluate the role of gut microbiota in diagnosis, treatment and prognosis of CCA and gallbladder carcinoma. The following research questions were posed before starting database research:

Do patients with CCA or gallbladder carcinoma show distinct gut microbiota changes compared to healthy controls?

Does the gut microbiota affect the postoperative outcome of CCA or gallbladder carcinoma patients who underwent curative-intent surgery?

Does the gut microbiota affect the chemotherapeutic or systemic treatment response of CCA or gallbladder carcinoma patients?

Does the gut microbiota affect the prognosis and the overall survival of patients suffering from CCA or gallbladder carcinoma?

## 3. Results

Systematic search yielded 154 results, of which 68 were not suitable for evaluation after screening of abstracts. Further 28 reports were excluded after full-text evaluation. Overall, a total of 12 studies and one systematic review were included. An overview of the search progress is shown in Figure 3. Detailed methods of included trials are shown in a supplementary file (see Appendix A).

### 3.1. Do Patients with CCA or Gallbladder Carcinoma Show Distinct Gut Microbiota Changes Compared to Healthy Controls?

In total, data from 282 CCA patients and 268 gallbladder carcinoma patients were evaluated. Two studies used cultivation methods for evaluation of single bacteria [43,44], and seven studies used novel sequencing techniques for evaluation of the entire gut bacteriota (six studies, [45,46,47,48,49,50]) or gut mycobiota (one study, [51]). Most of the studies excluded patients with other gastrointestinal diseases. The timing of measurement varied; four studies reported stool sampling before treatment, one reported mixed timing, and others did not specify the exact timing of measurements. Six studies excluded patients with prior use of antibiotics [46,47,48,49,50,51], one study stated the use of various antibiotics before stool sampling [45] and two studies did not report about prior application of antibiotics [43,44]. An overview of the characteristics of the eight included studies and one systematic review addressing the first research question is shown in Table 1 and Table 2.

#### 3.1.1. Gut Microbiota Changes in CCA Patients

The entire composition of gut microbiota of CCA patients compared to healthy controls was investigated in seven studies [45,46,47,48,49,50,51]. The results of diversity indices were inhomogeneous. One study reported higher α-diversity of CCA patients compared to healthy controls [48]. Two studies stated lower α-diversity of CCA patients compared to healthy controls (one of gut bacteriota [46] and one of gut mycobiota [51]). Three studies found no differences of α-diversity of CCA patients compared to healthy controls [47,49,50], and another study did not measure α-diversity of CCA patients compared to healthy controls [45]. The β-diversities within CCA patients were lower compared to healthy controls in one trial [48], whereas another trial did not report differences of β-diversities within CCA patients compared to healthy controls [45]. Further two trials reported a separated cluster of CCA patients based on β-diversities [46,47].

The composition of gut microbiota differed significantly between CCA patients and healthy controls to various degrees. A simplified overview of differences in gut microbial composition of CCA patients compared to healthy controls is shown in Figure 4.

The differences of gut mycobiota, which were evaluated in one trial by Zhang et al., are reported in Table 2. The authors found a higher abundance of opportunistic pathogenic fungi such as *Candida albicans* and a lower abundance of potential beneficial fungi such as *Saccharomyces cerevisiae* in CCA patients compared to healthy controls [51].

Besides comparisons with healthy controls, one study compared gut microbiota composition of CCA patients with patients suffering from primary sclerosing cholangitis (PSC) [45]. The authors reported no differences of diversity indices or taxa abundances between CCA patients and PSC patients with and without CCA.

Two of the publications suggested the use of predictive models to distinguish CCA patients from healthy subjects by evaluating gut microbiota. Zhang et al. suggested that the genera *Burkholderia*, *Caballeronia*, *Paraburkholderia*, *Faecalibacterium* and *Ruminococcus* were able to discriminate CCA patients from healthy controls [49]. Deng et al. postulated that the genera *Muribaculaceae unclassified*, *Lachnospiraceae NK4A136* group, *Escherichia/Shigella*, and *Klebsiella* might be promising biomarkers for distinguishing CCA patients from patients with hepatocellular carcinoma (HCC) or from healthy controls [47].

#### 3.1.2. Gut Microbiota Changes in Gallbladder Carcinoma

One systematic review and one case–control study were evaluated that address the gut microbial changes in gallbladder carcinoma patients. The studies focused on the impact of *Salmonella* spp. in the development of gallbladder carcinoma. There were no studies examining the composition of the entire gut microbiota of patients suffering from gallbladder carcinoma.

A systematic review published in 2014 by Nagaraja et al. emphasized the role of *Salmonella typhi* in development of gallbladder carcinoma [43]. The chronic carrier status as determined by cultivation was associated with gallbladder carcinomas, whereas the past medical history of infection with *Salmonella typhi* was not associated. A case–control study by Koshiol et al. compared stool samples from patients with gallbladder carcinoma (n = 13), patients with gallstone disease (n = 9) and healthy subjects (n = 13), but found no differences of *Salmonella* spp. by culture between the groups [44]. The authors added an additional systematic review with meta-analysis evaluating the associations between *Salmonella* spp. and gallbladder carcinoma, and concluded that the results support a potential role of *Salmonella* spp. measured by of stool culture in development of gallbladder carcinoma.

### 3.2. Does the Gut Microbiota Affect the Postoperative Outcome of CCA or Gallbladder Carcinoma Patients Who Underwent Curative-Intent Surgery?

None of the evaluated studies reported about the postoperative outcome of CCA patients or gallbladder carcinoma patients. Therefore, it is not possible to draw any conclusions regarding this research question.

### 3.3. Does the Gut Microbiota Affect the Chemotherapeutic or Systemic Treatment Response of CCA or Gallbladder Carcinoma Patients?

Two studies with a total of 44 patients evaluated the role of gut microbiota in response to programmed cell death receptor-1 (PD-1) antagonists in patients suffering from CCA [52,53]. Both studies reported the results of patients with advanced-stage disease after failure of first-line treatment. We found no reports evaluating the impact of gut microbiota on chemotherapeutic response of CCA or gallbladder carcinoma patients. An overview of the characteristics of the included studies is presented in Table 3 and Table 4.

The first study by Mao et al. reported on the response of patients with unresectable HCC and advanced biliary tract carcinoma [52]. The authors did not find differences in bacterial diversity indices or abundances between HCC patients and advanced biliary tract carcinoma patients and evaluated the treatment response of both entities together. In the group of patients with a complete or partial response or stable disease of ≤6 months, a higher abundance of phylum *Bacteroidota* and of order *Bacteroidales* and a lower abundance of order *Veillonellales* were found. Furthermore, the relative abundance of *Lachnospiraceae bacterium-GAM79* and a few species from the *Oscillospiraceae* family (*Ruminococcus callidus*, *Gemmiger formicilis*, *Eubacterium siraeum* and *Faecalibacterium genus*) were significantly enriched in patients with a complete or partial response or stable disease. The authors were able to perform a dynamic fecal sampling (sampling every three weeks before the anti-PD-1 infusion) in eight patients indicating the stability of gut microbiota along with the treatment in patients with a complete or partial response or stable disease.

Jin et al. compared gut microbiota of advanced-stage CCA patients with rapid progress within 6 months to advanced stage CCA patients with a slower progress [53]. The authors reported a higher abundance of phylum *Pseudomonadota* in patients with a rapid progress. A total of 26 *Pseudomonadota* species differed significantly between patients with rapid progress and patients with slower progress. Of particular interest were *Serratia Marcescens* and *Raoultella Planticola*, and the authors proposed these species as biomarkers of treatment response.

### 3.4. Does the Gut Microbiota Affect the Prognosis and the Overall Survival of Patients Suffering from CCA or Gallbladder Carcinoma?

Five studies (four human studies and one animal trial) reported on potential prognostic factors of the gut microbiota for promotion or progression of CCA or gallbladder carcinoma. Four studies evaluated CCA patients and one study reported on gallbladder carcinoma patients. The aims of these studies varied as two studies reported on gut microbial alteration in patients with advanced disease compared to patients with limited disease [48,51], and another study evaluated the association between the gut microbiota and the progression-free survival (PFS) and overall survival (OS) of advanced disease patients being treated with programmed cell death receptor-1 [52]. The animal trial focused on the development of CCA in PSC [54], and the last study evaluated epidemiological data regarding the mortality of *Salmonella typhi* or *Salmonella paratyphi* carriers [55]. An overview of the included studies is shown in Table 5 and Table 6. Further results of four of the five studies were previously mentioned in other subheadings.

Jia et al. (see also Section 3.1) found higher abundances of family *Oscillospiraceae* and lower abundances of family *Eubacteriaceae* as well as of genera *Allobaculum*, *Pediococcus*, *Pseudoramibacter*, and *Peptostreptococcus* in CCA patients with venous infiltration compared to CCA patients without venous infiltration [48]. Vascular infiltration is known to worsen the outcome of CCA patients [56,57].

Zhang et al. (see also Section 3.1) showed distinct differences of the gut mycobiota composition of patients with advanced-stage CCA (stage III and IV) compared to CCA patients with lower stages (stages I and II) [51]. The authors postulated a potential role of *Candida albicans* in advanced disease.

Mao et al. (see also Section 3.3) investigated the association between the gut microbiota and the oncological outcome of unresectable HCC and advanced biliary tract carcinoma [52]. Survival analysis revealed longer progression-free survival (PFS) and overall survival (OS) in patients with a higher abundance of species *Lachnospiraceae bacterium-GAM79*, *Erysipelotrichaceae bacterium-GAM147*, *Ruminococcus callidus*, *Alistipes megaguti* and *Bacteroides zoogleoformans*. Higher abundance of family *Veillonellaceae* was negatively associated with the patients’ PFS and OS. The authors separated biliary tract cancer patients from HCC patients and reported that biliary tract cancer patients with a higher abundance of order *Bacteroidales* had a significantly better PFS and OS. As mentioned before in the overall group analysis, a higher abundance of family *Veillonellaceae* was also negatively associated with the PFS and OS of biliary tract cancer patients.

Regarding the development of CCA, an animal trial by Zhang et al. emphasized the role of Gram-negative bacteria in CCA progression of patients suffering from PSC [54]. The authors treated mice with bile duct ligation (BDL) and mice with dextran sulfate sodium (DSS)-induced colitis with neomycin to eliminate Gram-negative bacteria, resulting in lower rates of CCA. After fecal transplantation of vancomycin-treated mouse stool in germ-free mice, a higher tumor burden of intrahepatic injected RIL-175 tumor cells (a murine HCC cell line) was found. The results could not be reproduced in mice free of colitis or PSC-like lesions.

The study published by Caygill et al. (see also Section 3.1) almost 30 years ago postulated that carriers of *Salmonella typhi* or *Salmonella paratyphi* have a large excess of cancer mortality compared to non-carriers, particularly of gallbladder carcinoma [55].

## 4. Discussion

In summary, the studies discussed above highlight a possible role of the gut microbiota in diagnosis, therapy and prognosis of CCA and gallbladder carcinoma.

### 4.1. Limitations of the Included Studies

Regarding the methodology of the studies evaluated, most studies had small sample sizes (on average, 29 CCA patients per study), and proper statistical planning and power analyses were missing. Differences in sequencing techniques and primers made the studies considered hardly comparable. In most of the included studies, only one stool sample was examined, which makes it difficult to draw conclusions due to the dynamic microbial environment, which is affected by the circadian rhythm and other external factors [58].

It is necessary to note that the included publications showed demographic differences of patients. Both studies about the clinical response to programmed cell death receptor-1 treatment provided just limited or no information about further clinical data of patients [52,53]. In four publications, patients suffering from CCA were significantly older and more often male than controls [45,46,49,50]. It is known that age and sex can affect the composition of the gut microbiota [59]. Furthermore, all authors said nothing about the diet of included patients, although food is one of the main modulators of the gut microbiota [60,61].

The majority of studies were conducted in China. Although it can be assumed that the included patients might have similar nutritional and lifestyle habits due to their origin, we found no consistent or overlapping observations between the studies. In general, the results are hardly comparable to those of patients from Western countries as it is known that origin and ethnicity affect the gut microbiota composition [62,63,64].

Most of the studies controlled for application of antibiotics as a major influencing factor of the gut microbial composition. The results of the study by Miyabe et al. are of limited value as different antibiotic therapies have been administered [45]. Due to microbial selection by antibiotics, it is clearly comprehensible that CCA patients with an antibiotic treatment are not comparable to other subjects without an antibiotic treatment.

Mao et al. analyzed data of patients suffering from HCC together with data of CCA patients, which is debatable [52]. Both tumor entities are primary liver cancers, but etiology, treatment modalities, tumor biology, prognosis and at last affected patient demographic data might differ significantly [65,66]. All of the studies refrained from distinguishing between the different sites of biliary tract cancer, although, as mentioned in the introduction, it is known that the location of the tumor has an influence on therapy and prognosis [11].

### 4.2. Alteration of Gut Microorganisms in CCA

The lack of clear and consistent changes in the microbial diversity indices is interesting, but unsurprising. Despite changes in habits or lifestyle, research indicates high stability of gut microbiota diversity, and an alteration of individual species might not be able to affect the diversity of the entire microbial community [64,67,68]. Noteworthy, the results of our review did not reveal a clear trend towards a lower abundance of health-promoting genera and a higher abundance of potential harmful species in CCA patients. Due to the inconsistent results, it can be hypothesized that the crucial factor for promotion or progression of CCA and gallbladder carcinoma may not be the species, but the metabolites that they provide. The liver is exposed to a variety of bacterial components and metabolites by the enterohepatic circulation, which contribute to the normal function of the human body (Figure 2) [69,70]. However, some of these molecules such as deoxycholic acid can also be harmful [39]. Usually, none of these molecules are exclusively synthesized by one single species. However, some of the possible harmful bacterial metabolites are also useful and contribute to the self-regulating and self-controlling equilibrium of the commensal bacteria [71].

Some of the altered genera observed in this review are discussed in more detail below.

#### 4.2.1. Genera with Lower Abundances in CCA

The included studies reported lower abundances of genera *Faecalibacterium*, *Ruminococcus*, *Megamonas*, *Burkholderia*, *Caballeronia* and *Paraburkholderia*.

The most commonly mentioned species of *Faecalibacterium* is *Faecalibacterium prausnitzii*, which is able to produce butyrate and other short-chain fatty acids. It is often considered an indicator for health due to its anti-inflammatory activity [72]. Another potentially harmful species of the same family belonging to the genus *Ruminococcus*, is *Ruminococcus gnavus*, which was found to be associated with inflammatory bowel disease and negative cardiovascular health indices [73,74]. Zhai et al. postulated that diarrhea-predominant irritable bowel syndrome might be provoked by *Ruminococcus*-driven stimulation of the serotonin biosynthesis [75]. However, *Ruminococcus* is also able to produce butyrate and other short-chain fatty acids [76].

*Megamonas* appears to be an indicator for the nutritional status of a patient, as studies indicate a higher abundance of *Megamonas* in obese subjects and a lower abundance in cancer patients suffering from cachexia [77,78]. It remains unclear whether *Megamonas* might be a sign of a cancer-related cachexia in the patients included in our review. Additionally, *Megamonas* was found to be negatively correlated with flavonoids, which are discussed to reduce cellular stress [79,80].

*Burkholderia* encompasses multiple pathogenic species involved in chronic infections [81]. The role of *Caballeronia* and *Paraburkholderia* for human health appears to be not clarified at the moment. Liu et al. found higher abundances of *Burkholderia*, *Caballeronia* and *Paraburkholderia* in Uyghur Chinese patients with ulcerative colitis compared to Han Chinese patients with ulcerative colitis [82]. The clinical consequence of this observation remains unclear.

#### 4.2.2. Genera with Higher Abundances in CCA

In the evaluated studies, higher abundances of genera *Actinomyces*, *Alloscardovia*, *Lactobacillus*, *Bacteroides*, *Alistipes*, *Shigella* and *Klebsiella* were reported in CCA patients.

Species of *Actinomyces* are known to cause a slowly progressing granulomatous disease named actinomycosis [83]. The infection is endogenous as the natural habitat of *Actinomyces* is the oral cavity. In addition to actinomycosis, species of *Actinomyces* can also lead to brain abscesses, or infections of others sites of the body [83]. Although it appears plausible that *Actinomyces* might be harmful for patients during chemotherapy due to the immunosuppressive situation, there is no evidence for a higher infection risk in chemotherapy-treated patients. Similar to *Actinomyces*, *Alloscardovia omnicolens*, a species belonging to the genus *Alloscardovia*, is sometimes found in infections of poor-conditioned patients [84]. Both genera might be a sign of the poor condition of the included patients.

Interestingly, CCA patients had higher abundances of *Lactobacillus*, a mutualistic genus that is thought to protect the balance of the human gut microbiota and promote human health [85,86]. One species of *Lactobacillus*, *Lactobacillus plantarum* provides indole-3-lactic acid, which ameliorates tumor growth and intestinal inflammation in colorectal cancer mice [87]. Yu et al. reported that *Lactobacillus lactis* is able to inhibit the non-alcoholic fatty liver disease progression by resorption of its metabolites via the gut–liver axis [88].

The genus *Bacteroides* is a highly relevant group for health and disease in humans and focus of several research projects. *Bacteroides* spp. are opportunistic human pathogens causing severe infections and abscesses and being resistant to a variety of antibiotics [89,90]. However, *Bacteroides* spp. are also mutualistic species protecting the equilibrium of the human gut microbiota from pathogens. *Bacteroides* spp. are discussed to be able to modulate the immune system, to enhance phagocytosis, to prohibit the colonization of pathogenic species, and to induce the colonization of beneficial species [91]. Animal-based diets and high-fat diets are associated with a higher abundance of *Bacteroides* spp. [67,92,93]. In mice, *Bacteroides* spp. is reported to show strong correlations with deoxycholic acid [94]. *Bacteroides fragilis* belonging to the genus *Bacteroides* is discussed to be a colorectal cancer-promoting species and a contributor to metastatic disease due to its modulation of cellular adhesion and epithelial tight junctions [95,96,97]. Overall, *Bacteroides* appears to be a key player, but its role in CCA needs to be further elucidated.

Another interesting and potentially health-promoting genus, which showed a higher abundance in the included CCA patients, is *Alistipes*. Recent research indicates a lower abundance of *Alistipes* in liver diseases, accompanied by an increasing reduction in *Alistipes* in the case of liver disease progression [98]. The higher abundance of this genus in CCA patients is surprising, but the results of Mao et al. underline the impact of *Alistipes* in these patients as a higher abundance of *Alistipes megaguti* was associated with a better survival [52]. Nevertheless, *Alistipes* spp. are associated with mental diseases such as anxiety and eating disorders. It is suggested that *Alistipes* spp. decreases serotonin by metabolism of the serotonin precursor tryptophan [98,99]. This observation is of interest, as cancer patients have a high risk for developing depression, which is known to affect patient survival [100].

*Klebsiella pneumoniae*, an ethanol-producing species, is discussed to promote non-alcoholic fatty liver disease due to its metabolites [101]. Species of the genera *Klebsiella* and *Shigella* are known to cause severe and potentially life-threatening diseases in humans, but both are also inhabitants of the gut microbiota.

### 4.3. Clinical Consequences and Future Aspects

Overall, we observed a deficiency in validated and consistent findings across multiple studies and, therefore, we see a lack of high-quality evidence. This lack of consistency in reported bacterial changes, without a clear trend towards either more harmful or more beneficial genera in patients with CCA, along with divergent results observed in different studies, aligns with the methodological inconsistencies. However, gut microbiota research is challenging due to the nature of the gut microbiota. It requires careful research planning that takes into account possible influencing factors and the microbial dynamics. Due to the limitations mentioned above, the analysis of gut microbiota in CCA patients and the usage of predictive models to distinguish between CCA patients and healthy controls is experimental and cannot be recommended at this time. More consistent and evident results might be achieved by larger samples-sizes and by considering influencing factors such as application of antibiotics, age, sex, nutrition, origin and ethnicity in future gut microbiota research. Due to the nature of gut microbiota and its circadian rhythm more longitudinal trials are needed to provide valid results. Experimental studies could investigate the effects of bacterial components and metabolites on the growth of CCA cells. Since our review emphasizes that high-quality data regarding the association between gut microbiota and BTC are widely lacking, both animal and human studies are of future interest.

## 5. Conclusions

Although evaluated studies show interesting alterations in intestinal microorganisms of patients suffering from BTC, the results are currently still limited by the methodological shortcomings and inconsistent findings mentioned above. Thus, no recommendation can be made at this time to evaluate the gut microbiota in patients with BTC regularly. Future studies need to address these shortcomings in order to clarify whether gut microbiota have an impact on the diagnosis, therapy and prognosis of patients with CCA and gallbladder carcinoma.

## Figures and Tables

**Figure 1 microorganisms-11-02363-f001:**
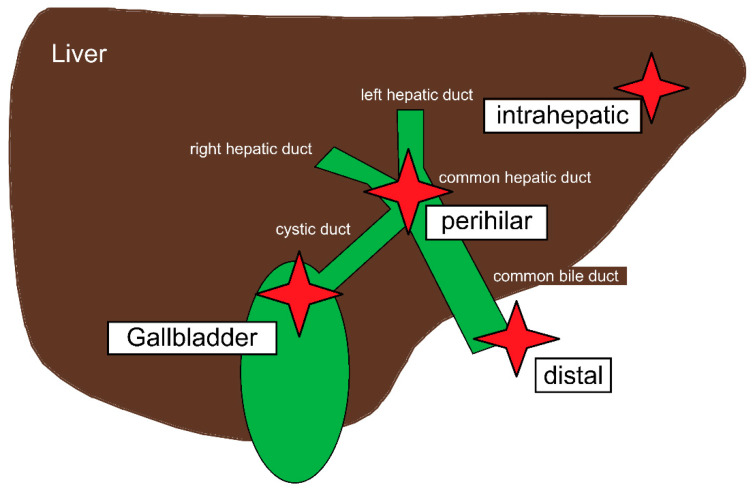
Systematics of biliary tract cancers differentiated in intrahepatic, perihilar (also known as Klatskin) and distal cholangiocarcinoma, and gallbladder carcinoma.

**Figure 2 microorganisms-11-02363-f002:**
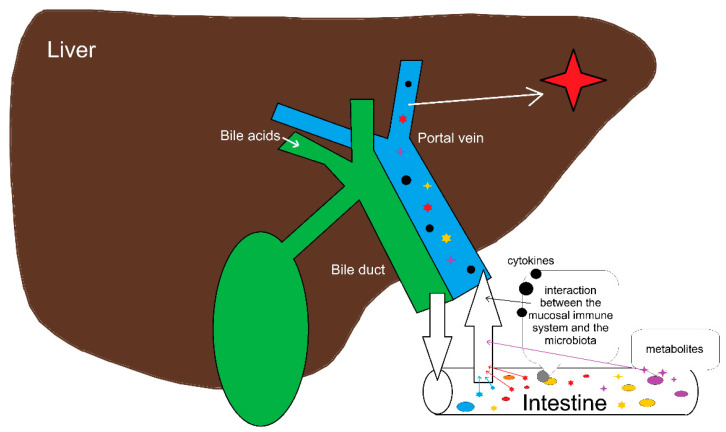
Systematics of the gut–liver axis: Bile acids and other molecules secreted by the liver shape the microbial milieu. The gut microbiota delivers cytokines, metabolites and other nutrient molecules via the portal vein.

**Figure 3 microorganisms-11-02363-f003:**
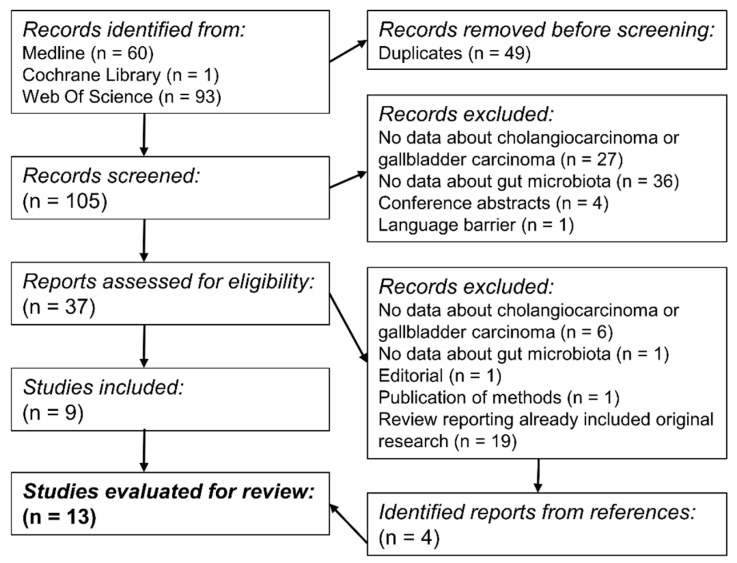
Overview of screening and assessing process. Thirteen reports (twelve trials and one systematic review) were evaluated for review.

**Figure 4 microorganisms-11-02363-f004:**
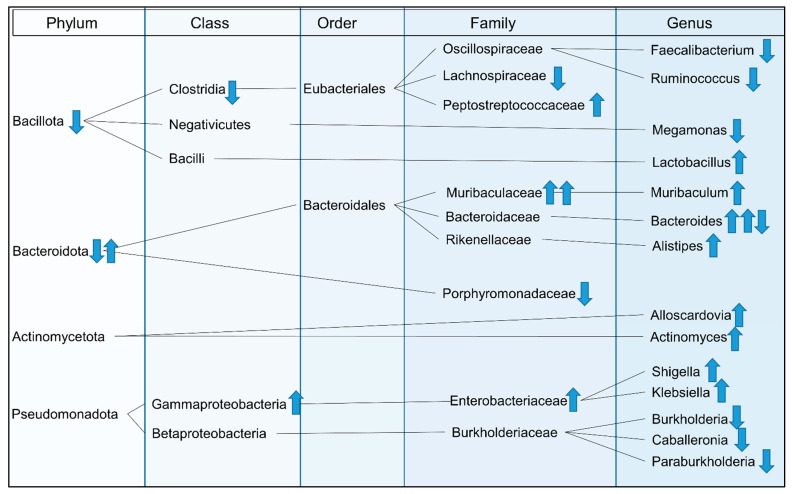
Simplified overview of gut microbial differences between CCA patients and healthy controls reported by the included trials. Arrows show an increase or decrease in the respective taxon of CCA patients compared to healthy controls. In case of more than one arrow, the taxon was reported in more than one publication.

**Table 1 microorganisms-11-02363-t001:** Overview of included original research addressing the first research question “Do patients with cholangiocarcinoma or gallbladder carcinoma show distinct gut microbiota changes compared to healthy controls?”—General information is sorted by year of publication (aCCA = all sites of cholangiocarcinoma or not specified in detail, HCC = hepatocellular carcinoma, iCCA = intrahepatic cholangiocarcinoma, n. s. = not specified, pCCA = perihilar cholangiocarcinoma, PSC = primary sclerosing cholangitis).

Reference and Author	Year	Country	Type of Study	N	Groups	Excluded
[43] Nagaraja et al.	2014	Australia	systematic review	255/861	*Salmonella typhi* carrier/non-carrier	n. s.
[44] Koshiol et al.	2016	Chile	case–control	13/9/13	gallbladder cancer/cholelithiasis/healthy	n. s.
[48] Jia/Lu/Zeng et al.	2019	China	case–control	28/28/16/12	iCCA/HCC/cirrhosis/healthy	metastatic CCA, mixed-type CCA
[49] Zhang et al.	2021	China	case–control	53/47/40	aCCA/cholelithiasis/healthy	other gastrointestinal or oncological diseases
[47] Deng/Li et al.	2022	China	case–control	46/143/40	aCCA/HCC/healthy	other gastrointestinal or oncological diseases, age > 80 years
[50] Ito et al.	2022	Japan	case–control	30/11/10	aCCA/BBD */healthy	cholangitis, severe medical comorbidities, previous treatment ^+^
[46] Ma et al.	2023	China	case–control	63/184/40	iCCA/HCC/healthy	other gastrointestinal or oncological diseases
[45] Miyabe et al.	2023	USA	case–control	11/16/31	pCCA ^#^/PSC+CCA/PSC	unsuccessful cannulation of bile duct, no bile duct sample, abnormal postsurgical anatomy
[51] Zhang et al.	2023	China	case–control	23/17	iCCA/healthy	mixed-type CCA, infectious diseases, other gastrointestinal, autoimmune or oncological diseases

* benign biliary tract disease (cholelithiasis or gallbladder polyps); ^+^ including also endoscopic retrograde cholangiography, ^#^ not specified in detail, more than 90% pCCA.

**Table 2 microorganisms-11-02363-t002:** Overview of included original research addressing the first research question “Do patients with cholangiocarcinoma or gallbladder carcinoma show distinct gut microbiota changes compared to healthy controls?”—Results are sorted by year of publication (see Table 1) (n. s. = not specified, Ref. = Reference).

Ref.	Method	Time Point	Antibiotics or Probiotics	α-Diversity *	Abundance *
[43]	cultivation	n. s.	n. s.	n. s.	*Salmonella typhi* carrier status was associated with gallbladder carcinoma
[44]	cultivation	n. s.	n. s.	n. s.	no detection of *Salmonella* spp. (neither in gallbladder cancer patients nor in controls)
[48]	16s RNA (V4)	n. s.	recently none	↑	↑ family *Peptostreptococcaceae* ↑ genera *Actinomyces*, *Lactobacillus*, *Alloscardovia*
[49]	16s rDNA (V3-V4)	before treatment	none for at least 2 months	no difference	↑ family *Muribaculaceae* ↑ genera *Bacteroides*, *Muribaculum*, *Alistipes* ↓ genera *Burkholderia*, *Caballeronia*, *Paraburkholderia*, *Faecalibacterium*, *Ruminococcus* (suggested biomarkers for differentiation between CCA patients and healthy controls)
[47]	16s rRNA (V3-V4)	n. s.	none for at least 8 weeks	no difference	↑ phylum *Bacteroidota*, family *Muribaculaceae* ↑ genera *Bacteroides*, *Shigella*, *Klebsiella*, *unclassified Lachnospiraceae NK4A136 group* ↓ phylum *Bacillota*, genus *Megamonas*
[50]	16s rRNA (V3-V4)	before treatment	none for at least 8 weeks	no difference	↑ class *Gammaproteobacteria* (main family *Enterobacteriaceae*) ↓ class *Clostridia* (main family *Lachnospiracea*) genera *Faecalibacterium* and *Coprococcus* enriched in healthy controls
[46]	16s rRNA (V3-V4)	before treatment	none for at least 2 months	↓	↓ phylum *Bacteroidota*, family *Porphyromonadaceae*
[45]	16S rRNA (n. s.)	Mixed	different antibiotics	n. s. ^+^	n. s. ^+^
[51]	ITS2 rDNA	before treatment	none for at least 3 months	↓	↑ phylum *Ascomycota*, genus *Candida* (main species *Candida albicans*), genus *Monographella* (main species *Monographella nivalis*) ↓ phylum *Mucoromycota*, phylum *Basidiomycota*, genera *Saccharomyces* (main species *Saccharomyces cerevisiae*), *Pichia* (main species *Pichia mandshurica*), *Mucor* (main species *Mucor circinelloides*), *Staphylotricum* (main species *Staphylotricum coccospurum*), *Actinomucor* (main species *Actinomucor elegans*), *Alternaria* (main species *Alternaria alternata*), *Fusarium* (main species *Fusarium oxysporum*), *Humicola* (main species *Humicola fuscoatra*)

* of cholangiocarcinoma patients compared to healthy subjects; ^+^ no comparison to healthy controls, no difference between other groups.

**Table 3 microorganisms-11-02363-t003:** Overview of included original research addressing the third research question “Does the gut microbiota affect the chemotherapeutic or systemic treatment response of CCA or gallbladder carcinoma patients?”—General information is sorted by year of publication (BTC = biliary tract cancer, HCC = hepatocellular carcinoma, n. s. = not specified).

Reference and Author	Year	Country	Type of Study	n	Groups	Excluded
[52] Mao/Wang/Long/Yang et al.	2021	China	cohort	30/35	advanced HCC/BTC ^+^	n. s.
[53] Jin et al.	2023	China	Phase II clinical trial	11 *	local advanced aCCA (clinical stage IV)	age < 75 years, cardiac or autoimmune disease, immunosuppressive treatment

* including 3 patients with a rapid progress and 8 patients with a progression-free survival of more than 6 months. ^+^ authors did not differentiate gut microbiota of HCC and aCCA patients.

**Table 4 microorganisms-11-02363-t004:** Overview of included original research addressing the third research question “Does the gut microbiota affect the chemotherapeutic or systemic treatment response of CCA or gallbladder carcinoma patients?”—Results are sorted by year of publication (see Table 3) (n. s. = not specified, Ref. = Reference).

Ref.	Treatment	Method	Time Point	Antibiotics or Probiotics	α-Diversity *	Abundance *
[52]	anti-PD-1 based systemic therapy	sequencing, n. s.	after failure of first-line therapy	none for at least 3 months	n. s.	↑ class *Negativicutes*, order *Enterobacterales*, order *Veillonellales*, family *Veillonellaceae* ↓ phylum *Bacteroidota*, order *Bacteroidales*
[53]	sintilimab plus anlotinib	16s rRNA (V3-V4)	after failure of first-line therapy	n. s.	n. s.	↑ phylum *Pseudomonadota* suggested species for response: *Serratia Marcescens* and *Raoultella Planticola*

* of patients with rapid progress (<6 months) compared to patients with slower progress.

**Table 5 microorganisms-11-02363-t005:** Overview of included original research addressing the fourth research question “Does the gut microbiota affect the prognosis and the overall survival of patients suffering from CCA or gallbladder carcinoma?”—General information is sorted by year of publication (BTC = biliary tract cancer patients, BDL mice = bile duct ligation mice, DSS-colitis mice = dextran sulfate sodium-induced colitis mice, HCC = hepatocellular carcinoma, iCCA = intrahepatic cholangiocarcinoma, n. s. = not specified).

Reference & Author	Year	Country	Type of Study	n	Groups	Excluded
[55] Caygill et al.	1994	UK	case–control	83/386	*Salmonella* carrier/non-carrier	subjects with ongoing infection
[48] Jia/Lu/Zeng et al.	2019	China	case–control	28/28/16/12	iCCA/HCC/cirrhosis/healthy	metastatic CCA, mixed-type CCA
[54] Zhang et al.	2021	USA	experimental (including mice)	n. s.	BDL mice/DSS-colitis mice/germ-free mice	-
[52] Mao/Wang/Long/Yang et al.	2021	China	cohort	30/35	advanced HCC/BTC ^+^	n. s.
[51] Zhang et al.	2023	China	case–control	23/17	iCCA/healthy	mixed-type CCA, infectious diseases, other gastrointestinal, autoimmune or oncological diseases

^+^ authors did not differentiate gut microbiota of HCC and aCCA patients.

**Table 6 microorganisms-11-02363-t006:** Overview of included original research addressing the fourth research question “Does the gut microbiota affect the prognosis and the overall survival of patients suffering from CCA or gallbladder carcinoma?”—Results are sorted by year of publication (see Table 5) (PFS = progression-free survival, OS = overall survival, Ref. = Reference, n. s. = not specified).

Ref.	Method	Time Point	Antibiotics or Probiotics	Results
[55]	cultivation	-	n. s.	carriers of *Salmonella typhi* and *Salmonella paratyphi* had a large excess of cancer mortality, particularly of gallbladder carcinoma (compared to non-carriers)
[48]	16s RNA (V4)	n. s.	recently none	in case of venous infiltration ↑ family *Oscillospiraceae* and ↓ family *Eubacteriaceae*, genera *Allobaculum*, *Pediococcus*, *Pseudoramibacter*, *Peptostreptococcus*
[54]	16s RNA (V4)	-	none	treatment with neomycin for elimination of Gram-negative bacteria resulted in fewer CCA, after dysbiotic fecal microbial transplantation germ-free mice developed liver myeloid cell accumulation, which is associated with worse outcome of CCA
[52]	sequencing, n. s.	n. s.	none for at least 3 months	↑ species *Lachnospiraceae bacterium-GAM79*, *Erysipelotrichaceae bacterium-GAM147*, *Ruminococcus callidus*, *Alistipes megaguti* and *Bacteroides zoogleoformans* => longer PFS and OS; ↑ family *Veillonellaceae* => shorter PFS and OS In biliary tract cancer patients: ↑ order *Bacteroidales* => longer PFS and OS, ↑ family *Veillonellaceae* = shorter PFS and OS
[51]	ITS2 rDNA	before treatment	none for at least 3 months	↑ *Candida* spp. (main species *Candida albicans*), *Dipodascus* spp., family *Ustilaginaceae*, family *Clavulinaceae*, and *Bipolaris* spp. CCA patients with stage III–IV compared to those with stage I–II; ↑ class *Sordariomycetes*, order *Xylariales*, family *Hyponectriaceae*, *Monographella* spp., *Annulohypoxylon* spp., increased in CCA patients with stage I–II compared to stage III–IV

## Data Availability

Not applicable.

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
