# Peer review of "Gut Microbiota in Diagnosis, Therapy and Prognosis of Cholangiocarcinoma and Gallbladder Carcinoma—A Scoping Review"

_microorganisms, 2023, doi:10.3390/microorganisms11092363_

Round 1

Reviewer 1 Report

Please add the experiments and improvements listed below to enhance the quality of the manuscript:

1-A well-organized manuscript with distinct section headers and subheadings is a must. Each part need to have a distinct goal and emphasis.

2-Eliminate any repetitions or contradictions you observe throughout the text. To make your views, use language that is clear and succinct.

3- To help in comprehending and graphically express difficult information, add good figures, diagrams, and charts. The Authors need to prepare a figure showing the role of gut microbiota in liver. In addition, please improve the quality of image in Figure 1.

4- Provide a more intriguing and informative introduction that emphasizes the significance of researching gut microbiota in patients with CCA and gallbladder cancer.

5-Provide additional information about the original study methodologies, including the detailed sequencing and culturing techniques employed to assess the gut microbiota.

6-To illustrate the selection, screening, and inclusion of studies, consider using a flowchart or graphic.

7-Provide the diversity indices wider context and highlight their importance in light of changes in the gut microbiota. To make the results section easier to read, consider combining similar findings.

8- In discussion, provide suggestions for further investigation based on the gaps identified in the current studies. This can include looking into the processes underlying the reported changes in the microbiota, assessing the potential treatment aspects, and doing longitudinal studies to monitor changes over time.

Be sure to thoroughly check the text for punctuation, grammatical, and spelling mistakes. Make sure the writing is precise and clear.

Author Response

Thank you for your careful review of our manuscript! We really appreciate your comments, which improved our manuscript.

Please add the experiments and improvements listed below to enhance the quality of the manuscript:

1-A well-organized manuscript with distinct section headers and subheadings is a must. Each part need to have a distinct goal and emphasis.

Response: The whole manuscript was revised. Similar to methodology and results, introduction and discussion are now also separated by subheadings.

2-Eliminate any repetitions or contradictions you observe throughout the text. To make your views, use language that is clear and succinct.

Response: The whole manuscript was revised to avoid repetitions and contradictions and to improve language. 

3- To help in comprehending and graphically express difficult information, add good figures, diagrams, and charts. The Authors need to prepare a figure showing the role of gut microbiota in liver. In addition, please improve the quality of image in Figure 1.

Response: Thank you for your suggestion! We added a new Figure called “Figure 2: Systematics of the "gut-liver axis": Bile acids and other molecules secreted by the liver shape the microbial milieu. The gut microbiota delivers cytokines, metabolites and other nutrient molecules via the portal vein” (page 3). The quality of Figure 1 was improved (page 2). 

4- Provide a more intriguing and informative introduction that emphasizes the significance of researching gut microbiota in patients with CCA and gallbladder cancer.

Response: We shortened and focused the introduction. In combination with the new Figure 2, the significance of researching gut microbiota should now be clearer.  

5-Provide additional information about the original study methodologies, including the detailed sequencing and culturing techniques employed to assess the gut microbiota.

Response: Due to the mass of methods, we have cited the original publications and prepared a supplementary file with all methods (see Supplementary Material - Methods). The respective regions of amplification were added to the tables.

6-To illustrate the selection, screening, and inclusion of studies, consider using a flowchart or graphic.

Response: Figure 2 (page 4) shows a flowchart illustrating the process of selection, screening and inclusion of studies.

7-Provide the diversity indices wider context and highlight their importance in light of changes in the gut microbiota. To make the results section easier to read, consider combining similar findings.

Response: Thank you for this comment, which addresses a difficult point of our work. We have revised the discussion in order to be able to discuss the problem of the inhomogeneous results in more depth (subheading 4.2, page 14). The results section has been proofread again. However, the results differ, a summary of the results combining similar findings is not possible in our eyes.

8- In discussion, provide suggestions for further investigation based on the gaps identified in the current studies. This can include looking into the processes underlying the reported changes in the microbiota, assessing the potential treatment aspects, and doing longitudinal studies to monitor changes over time.

Response: We added the subheading “Clinical consequences and future aspects” (page 16) to the discussion to contextualize the results from a clinical perspective. Moreover, we provide suggestions for future research.

Comments on the Quality of English Language

Be sure to thoroughly check the text for punctuation, grammatical, and spelling mistakes. Make sure the writing is precise and clear

Response: The whole manuscript was revised to improve punctuation and to eliminate grammatical and spelling mistakes.

Reviewer 2 Report

A good deal of work, but questions cannot be easily found or investigated.  I can't see where you could go next.  What specific investigations might give this field direction?

Author Response

Thank you for your careful review!

A good deal of work, but questions cannot be easily found or investigated.  I can't see where you could go next.  What specific investigations might give this field direction?

Response: Thank you for this important question! The entire manuscript was revised intensively. We revised also the discussion (page 14) and added the subheadings “Alteration of gut microorganisms in CCA” and “Clinical consequences” to give insights in possible interactions and to provide ideas for further research.  

Reviewer 3 Report

This is a compact review on CCA and intestinal microbiota showing no sufficient data to prove any correlation between them. Such negative data may not have big impact as a clinical interest, it is important to show such negative data. However, hypothesis on causal relationship between intestinal microbiota and CCA should be described in detail to develop further speculation. Metabolite of intestinal microorganism have to penetrate (absorb) through intestinal epithelium, and have to penetrate (secrete) hepatocyte. What kind of approaches do the authors speculate for further studies on the relationship between intestinal microbiota and CCAs? 

Author Response

Thank you for your careful review and your comments, which really improved our manuscript!

This is a compact review on CCA and intestinal microbiota showing no sufficient data to prove any correlation between them. Such negative data may not have big impact as a clinical interest, it is important to show such negative data. However, hypothesis on causal relationship between intestinal microbiota and CCA should be described in detail to develop further speculation. Metabolite of intestinal microorganism have to penetrate (absorb) through intestinal epithelium, and have to penetrate (secrete) hepatocyte. What kind of approaches do the authors speculate for further studies on the relationship between intestinal microbiota and CCAs?

Response: Thank you for your suggestions! We revised the introduction intensively adding also Figure 2, which illustrates the gut-liver axis. We also revised the discussion and added the subheadings “Alteration of gut microorganisms in CCA” and “Clinical consequences” to give insights in possible interactions and to provide ideas for further research.

Round 2

Reviewer 1 Report

All of the reviewers' comments were satisfactorily addressed by the authors.

no any

Reviewer 3 Report

Revisions are well described.